# Depth-dependent peridotite-melt interaction and the origin of variable silica in the cratonic mantle

Emma L. Tomlinson [1✉] & Balz S. Kamber [2]

Peridotites from the thick roots of Archaean cratons are known for their compositional diversity, whose origin remains debated. We report thermodynamic modelling results for reactions between peridotite and ascending mantle melts. Reaction between highly magnesian melt (komatiite) and peridotite leads to orthopyroxene crystallisation, yielding silica-rich harzburgite. By contrast, shallow basalt-peridotite reaction leads to olivine enrichment, producing magnesium-rich dunites that cannot be generated by simple melting. Komatiite is spatially and temporally associated with basalt within Archaean terranes indicating that modest-degree melting co-existed with advanced melting. We envisage a relatively cool mantle that experienced episodic hot upwellings, the two settings could have coexisted if roots of nascent cratons became locally strongly extended. Alternatively, deep refractory silica-rich residues could have been detached from shallower dunitic lithosphere prior to cratonic amalgamation. Regardless, the distinct Archaean melting-reaction environments collectively produced skewed and multi-modal olivine distributions in the cratonic lithosphere and bimodal mafic-ultramafic volcanism at surface.

[1] Department of Geology, Trinity College Dublin, Dublin, Dublin 2, Republic of Ireland. [2] School of Earth and Atmospheric Sciences, Queensland University of Technology, Brisbane, QLD, Australia. ✉email: tomlinse@tcd.ie

Continents are a hallmark feature of the Earth and have been in existence for at least 80% of Earth's history. They contain a mere 0.35% of the total mass of the planet but play an essential role in many geo-bio-chemical cycles, including the carbon cycle. The c. 35 km thick continental crust is underlain by a much deeper (up to 250 km) sub-continental lithospheric mantle (SCLM). Together, the crust and SCLM form the buoyant continental plates, representing most of the emerged landmass on Earth, critical for relative sea level and the long-term climate of the planet.

By comparison with younger peridotite, Archaean (>2.5 Ga) rocks from the SCLM, are very rich in magnesium with typical olivine forsterite contents of 92–94%. The highly refractory composition of the SCLM is consistent with its formation by extensive (30–50%) melt extraction from fertile peridotite[1] during one or several melting events. Rhenium-Os depletion ages of peridotites and of sulphide inclusions in diamonds indicate melting was ancient, having occurred between 3.5 and 2.5 Ga[2,3] broadly coeval with cratonic crust formation. The refractory residue was comprehensively depleted in heat-producing elements, resulting in a cool and viscous SCLM that has remained tectonically stable and isolated from mantle convection ever since.

One unexpected feature of many peridotites from Archaean SCLM is that they have lower MgO/SiO$_2$ ratios, for a given magnesium-number (Mg#), than is observed in oceanic and off-craton continental peridotites. Mineralogically, this characteristic manifests as a high modal orthopyroxene abundance, first discussed by Boyd (1989)[4], who noted that low temperature peridotites from the Kaapvaal craton contain highly forsteritic olivine yet only 55–80% modal olivine abundance (Fig. 1). This "excess" silica relative to post-Archaean peridotites is most extreme in

Kaapvaal harzburgites, however, harzburgites with high ortho-pyroxene modes have since also been described from the Slave, Siberian, Wyoming and Tanzanian cratons. By contrast, the North Atlantic craton (NAC) is the archetype host of peridotites characterised by extreme modal olivine abundances of 80–100% for a similar magnesium-number, a feature that has been attributed to extensive partial melting leading to orthopyroxene exhaustion[5,6]. It is difficult to produce pure dunite by simple peridotite melting as it requires temperatures in excess of 250–300 °C above the solidus[7,8].

There is currently no consensus about the origin and significance of the low MgO/SiO$_2$ ratio of the dominant type of cratonic peridotite. If viewed as a primary feature, it could have resulted from melting of fertile upper mantle (pyrolite) at very high pressure[9,10] or melting of non-pyrolitic mantle that had previously undergone fractionation in a magma ocean[11] or in very large impact melts seas[12]. Alternatively, low MgO/SiO$_2$ ratios may be the result of secondary addition of silica to depleted peridotite. In models working with plate tectonic processes, silica enrichment has been viewed as a result of serpentinisation[13] or melt-rock reaction at ocean ridges[14], or fluid-fluxing of the mantle wedge above a subduction zone followed by crustal thickening[15–17]. In models that view the SCLM as the residue after melting by one or more hot mantle upwellings[10,18–20], silica enrichment has been attributed to interaction with melts derived from restitic[21] or subducted[22] eclogite. Clearly, these various scenarios have substantially different consequences for the geo-dynamic evolution of the Earth over time.

In this study, we compiled a comprehensive (n = 425) global data set of mineral modal abundances and major element compositions for Archaean cratonic peridotites (Table 1) and

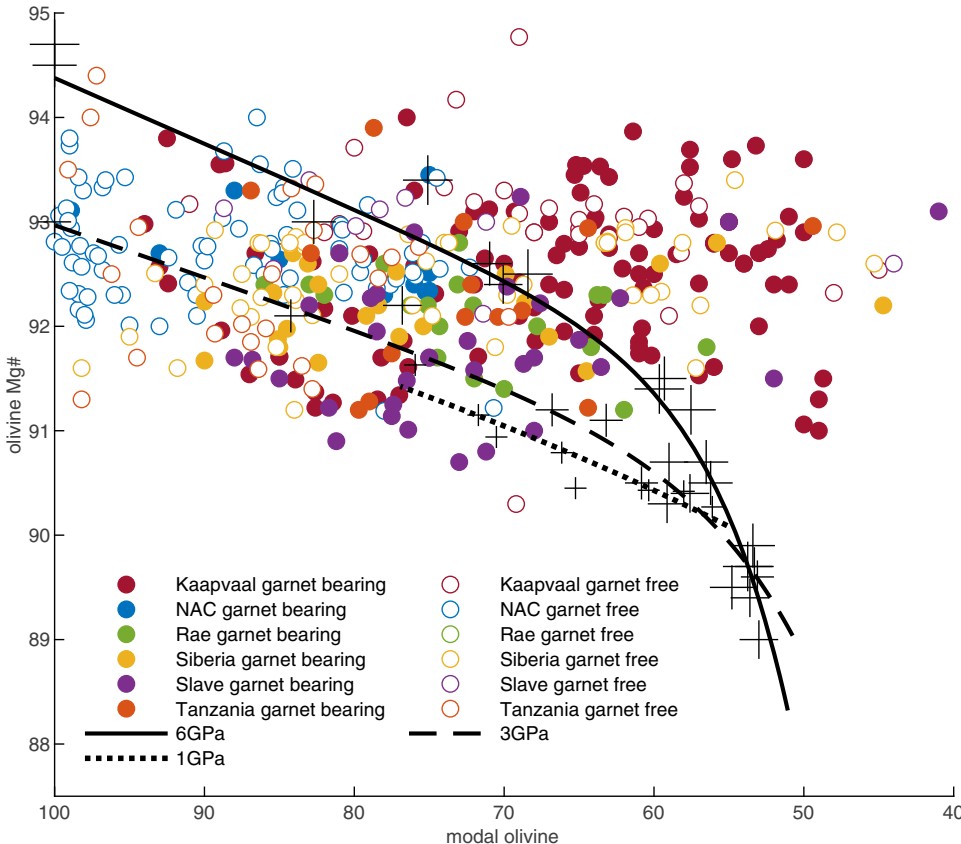

**Fig. 1 Modal olivine 'vs' olivine forsterite content for peridotites from the six studied cratons.** Also shown are the compositions (+ symbols, size scaled for pressure) and fitted trends for the evolution of experimental residues after melting of fertile peridotite at 1 GPa[43], 3 GPa and 6 GPa[41].

**Table 1 Summary of key information on the sub-cratonic lithosphere. Data is shown for individual locations with more than eight data points.**

| Craton | Re-Os TRD age (Ga) | Location | Region | Host magma | n | NiO Mean | WR Mg# Mean | ol Fo Mean | $Cr_2O_3/Al_2O_3$ Mean | Olivine mode Mean | Olivine mode Median | $MgO/SiO_2$ Mean | $MgO/SiO_2$ Median | KDE1 | % | KDE2 | % |
|---|---|---|---|---|---|---|---|---|---|---|---|---|---|---|---|---|---|
| **NAC** | 2.7–3.1 | **All** | | | **88** | **0.33** | **92.7** | **92.7** | **3.32** | **88.2** | **88.0** | **1.160** | **1.167** | **1.210** | **57** | **0.091** | **43** |
| | | **Garnet bearing** | | | **10** | **0.32** | **92.2** | **92.7** | **0.55** | **81.92** | **77.0** | **1.103** | **1.088** | **1.067** | **71** | **1.189** | **29** |
| | | **Garnet free** | | | **78** | **0.33** | **92.7** | **89.2** | **3.59** | **89.03** | **89.8** | **1.167** | **1.170** | **1.213** | **61** | **1.105** | **39** |
| | | Ubekendt Ejland | West Greenland | alkali basalt | 27 | 0.34 | 92.6 | 92.7 | 8.04 | 97.13 | 98.1 | 1.225 | 1.230 | 1.227 | 100 | – | – |
| | | Wiedemann Fjord | East Greenland | alkali basalt | 39 | 0.33 | 92.9 | 92.9 | 0.91 | 83.54 | 84.7 | 1.128 | 1.132 | 1.133 | 100 | – | – |
| | | Sarfartoq | Southwest Greenland | kimberlite | 20 | 0.30 | 92.2 | 92.6 | 0.60 | 86.24 | 85.5 | 1.146 | 1.145 | 1.196 | 57 | 1.088 | 43 |
| **Kaapvaal** | 2.8 | **All** | | | **139** | **0.30** | **92.6** | **92.6** | **0.42** | **67.30** | **65.3** | **0.991** | **0.977** | **1.115** | **30** | **0.959** | **70** |
| | | **Garnet bearing** | | | **108** | **0.30** | **92.5** | **92.5** | **0.34** | **66.57** | **65.0** | **0.987** | **0.971** | **1.098** | **33** | **0.949** | **67** |
| | | **Garnet free** | | | **31** | **0.30** | **92.8** | **92.9** | **0.71** | **67.69** | **66.7** | **1.002** | **1.002** | **1.160** | **26** | **0.983** | **74** |
| | | Finsch | Kimberley block | kimberlite | 33 | 0.31 | 92.2 | 92.1 | 0.46 | 77.75 | 81.4 | 1.052 | 1.062 | 1.116 | 58 | 0.958 | 42 |
| | | Kimberley | Kimberley block | kimberlite | 53 | 0.30 | 92.7 | 92.8 | 0.49 | 64.12 | 64.0 | 0.966 | 0.964 | 0.944 | 69 | 1.114 | 31 |
| | | Premier | Witwatersrand terrane | kimberlite | 8 | 0.28 | 91.5 | 92.0 | 0.29 | 64.75 | 64.0 | 0.957 | 0.957 | – | – | – | – |
| | | Jagersfontein | Witwatersrand terrane | kimberlite | 12 | 0.27 | 92.7 | 93.0 | 0.32 | 70.11 | 68.0 | 1.016 | 0.997 | – | – | – | – |
| | | Lesotho | Witwatersrand terrane | kimberlite | 29 | 0.30 | 92.8 | 92.9 | 0.35 | 62.79 | 62.4 | 0.953 | 0.953 | 0.957 | 100 | – | – |
| **Siberia** | 2.9 | **All** | | | **60** | **0.30** | **92.2** | **92.4** | **0.51** | **75.27** | **77.2** | **1.037** | **1.060** | **1.104** | **54** | **0.961** | **44** |
| | | **Garnet bearing** | | | **18** | **0.30** | **91.8** | **92.2** | **0.39** | **75.49** | **79.2** | **1.037** | **1.066** | **1.083** | **58** | **0.978** | **42** |
| | | **Garnet free** | | | **42** | **0.30** | **92.4** | **92.5** | **0.62** | **75.17** | **76.3** | **1.036** | **1.059** | **1.115** | **54** | **0.957** | **44** |
| | | Obnazhennaya | Northern Siberian | kimberlite | 8 | 0.29 | 91.2 | 92.1 | 0.42 | 70.40 | 70.6 | 0.937 | 0.931 | 1.109 | 54 | 0.973 | 46 |
| | | Udachnaya | Central Siberian | kimberlite | 52 | 0.30 | 92.4 | 92.5 | 0.55 | 77.04 | 82.5 | 1.052 | 1.078 | – | – | – | – |
| **Slave** | 2.6–2.75 | **All** | | | **50** | **0.30** | **91.9** | **92.1** | **0.55** | **73.24** | **75.0** | **1.013** | **1.015** | **0.970** | **50** | **1.071** | **50** |
| | | **Garnet bearing** | | | **41** | **0.30** | **91.6** | **91.8** | **0.46** | **72.60** | **74.3** | **1.002** | **1.005** | **0.981** | **66** | **1.069** | **34** |
| | | **Garnet free** | | | **9** | **0.29** | **92.9** | **92.9** | **0.97** | **75.95** | **78.3** | **1.060** | **1.086** | – | – | – | – |
| | | Jericho | Northern Slave | kimberlite | 17 | 0.29 | 91.7 | 92.2 | 0.42 | 71.98 | 70.5 | 0.998 | 0.991 | – | – | – | – |
| | | Lac de Gras | Central Slave | kimberlite | 33 | 0.30 | 92.0 | 92.0 | 0.62 | 73.78 | 76.5 | 1.020 | 1.027 | – | – | – | – |
| | | Gahcho Kué | Southern Slave | kimberlite | 18 | 0.31 | 91.9 | 91.9 | 0.65 | 76.82 | 77.5 | 1.036 | 1.026 | – | – | – | – |
| **Rae** | 2.75 | **All** | | | **50** | **0.31** | **91.4** | **92.1** | **0.45** | **73.44** | **74.2** | **1.016** | **1.029** | **1.052** | **73** | **0.943** | **27** |
| | | **Garnet bearing** | | | **45** | **0.31** | **91.4** | **83.7** | **0.38** | **73.70** | **74.4** | **1.015** | **1.033** | **1.042** | **75** | **0.934** | **25** |
| | | **Garnet free** | | | **5** | **0.31** | **91.8** | **0.0** | **1.05** | **71.15** | **71.9** | **1.024** | **1.024** | – | – | – | – |
| | | Somerset Island | Northern Rae | kimberlite | 50 | 0.31 | 91.4 | 92.1 | 0.45 | 73.44 | 74.2 | 1.016 | 1.029 | 1.039 | 80 | 0.930 | 20 |
| **Tanzania** | 2.5–2.9 | **All** | | | **38** | **0.31** | **92.5** | **92.4** | **1.38** | **82.79** | **83.9** | **1.105** | **1.107** | **1.200** | **49** | **1.100** | **51** |
| | | **Garnet bearing** | | | **14** | **0.26** | **92.4** | **92.4** | **0.42** | **73.45** | **73.7** | **1.030** | **1.030** | – | – | – | – |
| | | **Garnet free** | | | **24** | **0.32** | **92.5** | **92.5** | **1.93** | **88.24** | **88.6** | **1.148** | **1.135** | **1.209** | **38** | **1.108** | **62** |
| | | Labait | Eastern craton margin | melilitite | 8 | 0.30 | 92.2 | 92.4 | 0.75 | 81.24 | 78.0 | 1.100 | 1.096 | – | – | – | – |
| | | Lashaine | Eastern craton margin | Ankaramite, carbonatite | 26 | 0.30 | 92.3 | 92.3 | 0.82 | 81.03 | 83.3 | 1.086 | 1.096 | 1.136 | 70 | 0.996 | 30 |

Bold indicates summary data from all studied locations within the craton.

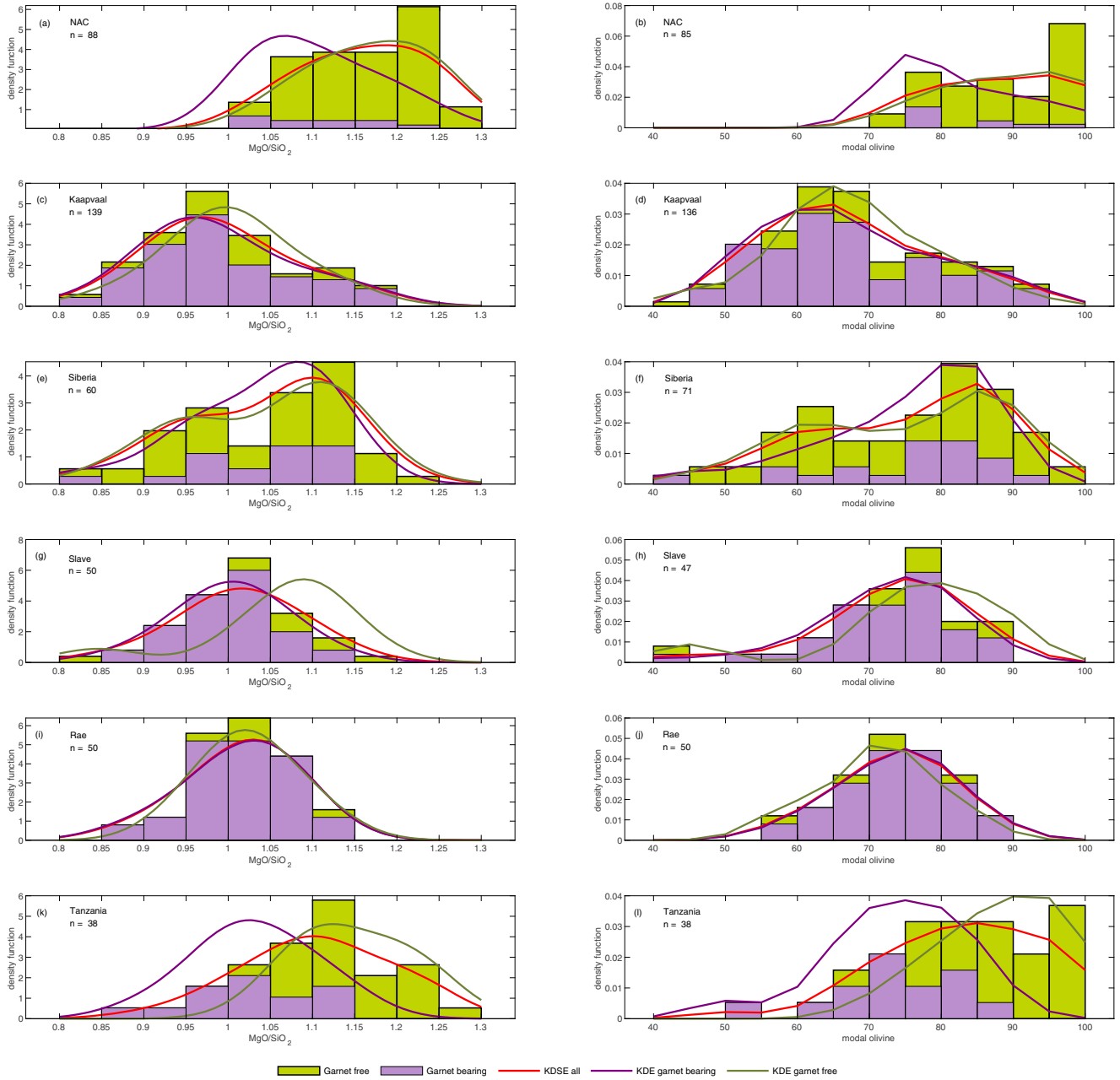

**Fig. 2 Normalised stacked histograms and Kernel Distribution Estimations for xenoliths from the studied Archaean cratons.** MgO/SiO2 (left, bandwidth = 0.05) and modal olivine (right, bandwidth = 5) in garnet-bearing and garnet-free peridotites from NAC (**a-b**), Kaapvaal (**c-d**), Siberia (**e-f**), Slave (**g-h**), Rae (**i-j**) and Tanzania (**k-l**). See supplementary data 1 for full dataset. Discrepancies between the number of samples plotted in MgO/SiO2 and modal olivine reflect differences in data types presented in different studies.

compared observed modes and compositions to those of modelled residues of high-pressure melting of fertile peridotite. We next used thermodynamic models to find petrologically permissible mechanisms for producing the highly variable orthopyroxene and olivine abundances in the SCLM. This approach is independent of assumptions about the prevailing tectonic regime and depth of melt extraction and does not rely on the limited trace element partitioning data available for phases present during very high temperature and/or high-pressure melting.

## Results
### Global patterns of olivine abundance in cratonic peridotites.
The modal distributions of olivine abundance in the compiled cratonic xenolith populations were statistically assessed with

Kernel Density Estimation (KDE). To facilitate comparisons with experimental and modelled residues produced at temperatures above the solidus, we also assessed the modal distribution of MgO/SiO$_2$. This is necessary because the mineral abundances within a given peridotite composition are dependent on the prevailing pressure and temperature of residence. The KDE plots (Fig. 2) show that the modal olivine contents in, and MgO/SiO$_2$ compositions of, cratonic peridotites are not normally distributed. Rather, distributions are either skewed or broad/multimodal both within single cratons and at individual sample locations, as previously noted[13]. By contrast, the Mg# and NiO contents of cratonic peridotite and olivine are typically normally distributed and are not correlated with olivine abundance, contrary to what is expected for peridotite melting (Supplementary figure S2).

The studied cratonic peridotites are characterised by three main modes of olivine; low (63–68%), medium (76–82%), and high (>93%), with low olivine peridotites having complementary high modal orthopyroxene. In the following, the three types are termed "orthopyroxene-rich", "normal" and "olivine-rich" peridotite, respectively. Normal peridotites dominate the populations of garnet-bearing and garnet-free peridotite xenoliths from the Slave, Rae and Siberian cratons and garnet-bearing peridotites from Tanzania.

Orthopyroxene-rich peridotites are best represented on the Kaapvaal craton, which has a KDE peak at 65% olivine ($MgO/SiO_2$ = 0.96) and a positive skew towards more normal olivine abundances. However, there is regional variability in olivine and orthopyroxene content within the Kaapvaal craton, as demonstrated by the peridotite population from Finsch, which has normal olivine abundances and a negative skew towards more orthopyroxene-rich compositions. Orthopyroxene-rich peridotites are also an important component of the peridotite xenolith population at Udachnaya (Siberia), where the olivine distribution is bimodal with peaks at 64% olivine (43% of population) and 83% olivine (57% of population). Peridotites with >30% orthopyroxene have also been described from the Slave (Torrie and Lac de Gras[23,24]) and Rae (Somerset Island[25]) cratons, where orthopyroxene-rich peridotites form a subordinate population, causing a negative skew away from the KDE peak at normal olivine abundance. This clearly shows that excess silica is not unique to the Kaapvaal SCLM.

Olivine-rich peridotites are exemplified by peridotites from the NAC. Dunites are abundant among spinel-bearing peridotites from Wiedemann Fjord[5] and Ubekendt Ejland[26] and garnet-free and shallow garnet-bearing peridotites from Sarfartoq[27,28] and Pyramidefjeld[29], which show a broad KDE profile with a peak at 94.5% olivine ($MgO/SiO_2$ = 1.21) and a negative skew towards lower olivine abundances. Dunites also form a subordinate population among peridotites from Udachnaya and Obnazhennaya in Siberia[30–32] and at Lashaine and Olmani in Tanzania[33–36]. These coarse, Mg-rich dunites often coexist with Fe-rich (Fo < 90) and generally finer grained and deformed dunites that have been interpreted as either cumulates from primitive melts[28] or products of melt-rock reaction leading to Fe-enrichment[37].

The available data indicate possible stratification of the cratonic lithosphere in some locations, with olivine-rich peridotite at shallow depth[38] and normal and/or orthopyroxene-rich peridotite at greater depth, although unavoidable sampling bias could distort the observed distributions. Within the NAC, olivine-rich peridotites have been exhumed by alkali basalts at Ubekendt Ejland and Wiedemann Fjord and also occur among shallow (90–115 km) peridotite xenoliths from the Sarfartoq kimberlite dykes, whereas peridotites from the deeper mantle below Sarfartoq have normal olivine abundances[27,28]. A similar picture emerges from Tanzania, where olivine-rich peridotites are restricted to the shallow lithosphere (spinel facies) and normal peridotites occur at greater depth[39]. At Udachnaya, which has a bimodal distribution of olivine abundance, olivine-rich, and normal peridotites occur throughout the same depth range but define distinctive Re-Os age populations, suggesting that the two lithologies had different formation histories before being amalgamated during the final assembly of the Siberian craton[32]. Vertical and temporal stratification of peridotite types suggests that the skewed and multimodal distributions of olivine abundance are unlikely to reflect a continuum process but rather distinct events in the history of the cratonic lithosphere.

**Comparison of fertile mantle melting residues and natural peridotites**. Cratonic peridotites have been interpreted as

residues after high-degree melt extraction from fertile peridotite[1,4,10,40]. We have evaluated this possibility by comparing the compiled data to residues of pyrolite melting predicted by phase equilibria modelling undertaken using THERMOCALC at 1–6 GPa (supplementary information), model output is shown in Fig. 3. High-degree melting is easier to achieve at higher pressure, at 1 GPa 40% melting is achieved 250–300 °C above the solidus, but this drops to just 90–100 °C above the solidus at 4 GPa[7,8]. Our findings are in agreement with previous experimental[41,42] and theoretical work[7,15] showing that residues after high-pressure melting are characterised by higher $SiO_2$ contents than those at lower pressure. This is due to the orthopyroxene-forming peritectic reaction occurring in the melt field above 3 GPa[41,42] (Table 2, Fig. 4b). Our model analysis agrees with the well-established observation that the majority of peridotites from the Kaapvaal craton[4,7,10,41], as well as orthopyroxene-rich peridotites from the Siberian, Slave and Rae cratons, are too enriched in $SiO_2$ to be residues of melting of pyrolitic mantle, even at high pressure[7,41]. When compared to the modelled residue trends, orthopyroxene-rich peridotites from Kaapvaal and elsewhere typically plot at apparent pressures >>5 GPa in Mg# vs $SiO_2$-FeO, but at lower and variable apparent pressure in Mg# vs $Al_2O_3$ (Fig. 3a–c). Such contradictory pressure requirements were noted by Herzberg (2004), and are a further strong argument against simple melting of pyrolite mantle as the origin of orthopyroxene-rich peridotites.

By contrast, residues after low pressure melting have lower $SiO_2$ contents due to an olivine-forming peritectic reaction at <2 GPa[43,44] (Table 2, Fig. 4a). The compositions of cratonic dunites with high Mg# (>93, such as many from Wiedemann Fjord) are consistent with derivation by high-degree melting at or close to the orthopyroxene-out reaction[26,38]. Importantly, however, the majority of olivine-rich cratonic peridotite suites extend to compositions that are too $SiO_2$-poor (or too Fe-rich) to be residues of simple melting of pyrolitic mantle, even at low pressures of 1–2 GPa (Fig. 3a, c). Olivine-rich peridotites from the NAC and elsewhere also yield contradictory apparent pressure information, with impossibly low pressures of «1 GPa predicted by Mg# vs $SiO_2$-FeO. In particular, peridotites from Ubekendt Ejland (NAC) extend to high FeO at high MgO. Olivine-rich peridotites also extend to very high $Cr_2O_3$ and have $Cr_2O_3/Al_2O_3$ ratios that fall well outside of the calculated and experimental bounds for melting of fertile peridotite and are up to an-order-of-magnitude higher than other cratonic peridotites (Fig. 3e, f).

High $Cr_2O_3/Al_2O_3$ ratios of cratonic peridotites[45] and diamond inclusions[46] are commonly used to infer a low pressure melting origin for the SCLM. This is based on the observation that while the $Cr_2O_3$ content of the residue remains approximately constant, $Al_2O_3$ preferentially partitions into the liquid at low pressure. The highest partition coefficient, $BulkD_{Cr/Al}^{residue/liquid}$ is achieved at the solidus at 1–1.5GPa[45] and, therefore, low pressure fractional melting may produce residues with high $Cr_2O_3/Al_2O_3$. However, low $Al_2O_3$ may also reflect high-pressure melting at or close to garnet exhaustion: melting at higher pressure leads to a gradual increase in $BulkD_{Cr/Al}^{residue/liquid}$, while progressive melting within the spinel stability field leads to a sharp decrease in $BulkD_{Cr/Al}^{residue/liquid}$ to 10%, such that by 20% melting, the D value is similar at 1 GPa as at 3–4 GPa. Therefore, residues with similar $Cr_2O_3/Al_2O_3$ are produced across a range of pressures at the high degrees of melting expected for cratonic peridotites. Consequently, $Cr_2O_3/Al_2O_3$ cannot be used to distinguish between low- or high-pressure melting for the SCLM and both shallow and deep

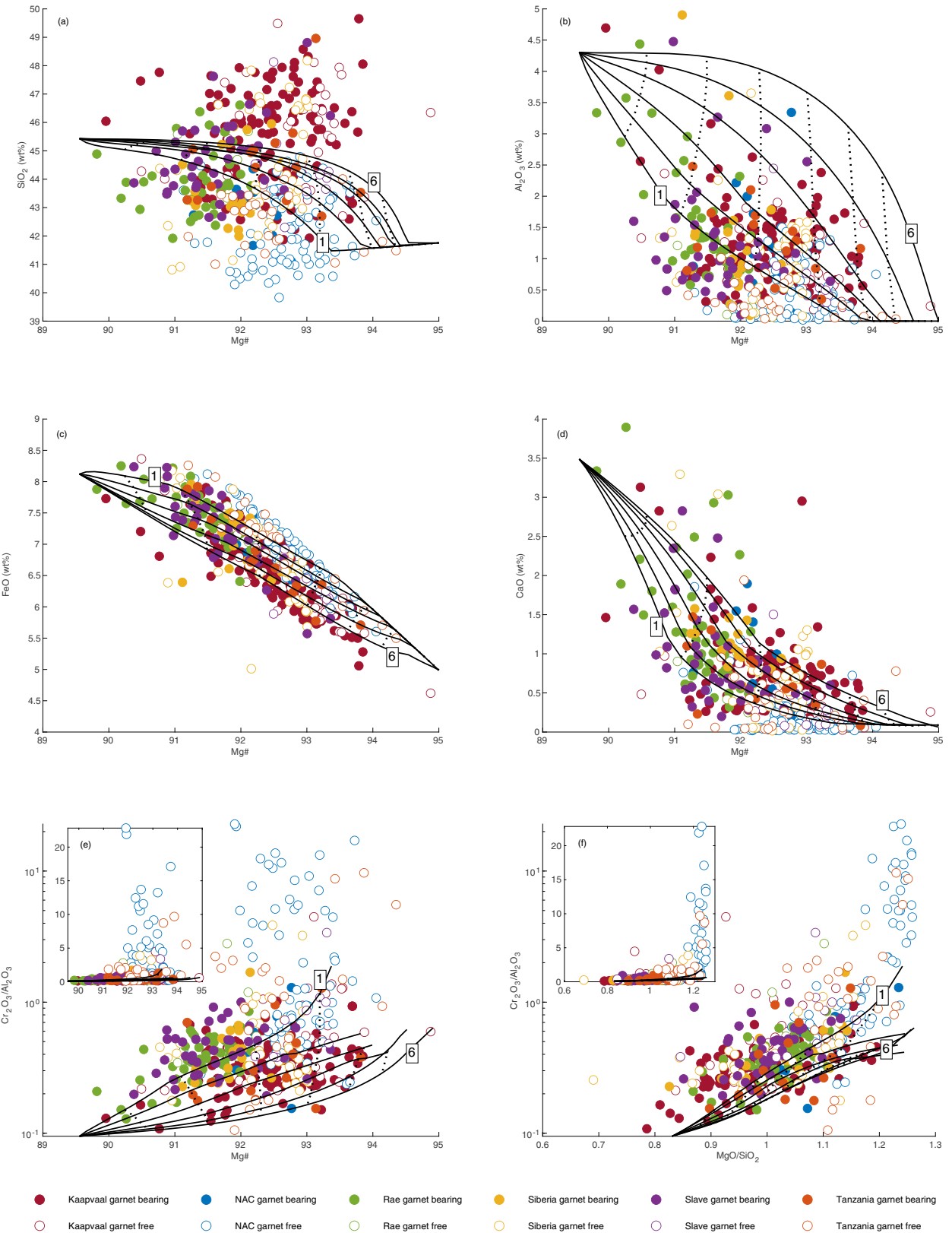

**Fig. 3 Bulk compositions of individual garnet bearing (solid symbols) and garnet free (open symbols) xenoliths from Archaean cratons compared to the modelled compositions of residues of melting of fertile peridotite KR4003.** (**a**) Mg# - SiO$_2$, (**b**) Mg# - Al$_2$O$_3$, (**c**) Mg# - FeO, (**d**) Mg# - CaO, (**e**) Mg# - Cr$_2$O$_3$/Al$_2$O$_3$, (**f**) MgO/SiO$_2$- Cr$_2$O$_3$/Al$_2$O$_3$. Pressure is shown in 1GPa increments from 1 to 6GPa (solid lines, low and high pressures are labelled) and degree of melting is marked in 10% increments (dotted lines). Note log scale in parts e and f, the insets show the same data on a linear scale.

**Table 2 Calculated reactions for melting of fertile peridotite and hybrid mantle. Note: reactions were not calculated for fields spanning <4% melting.**

| | KR4003 (fertile peridotite) | Komatiite + KR4003 (50:50) | Basalt + KR4003 (20:80) |
|---|---|---|---|
| 1 GPa | - | - | 1pl + 0.12cpx + 0.48ol = 0.54opx + 1liq |
| | **1.06cpx + 0.03opx = 0.20ol = 1liq** | **1.06cpx + 0.03opx = 0.18ol = 1liq** | **1.04cpx + 0.05 opx = 0.19ol = 1liq** |
| | **1.04opx = 0.04ol + 1liq** | **1.03opx = 0.03ol + 1liq** | **1.08opx = 0.07ol + 1liq** |
| 2 GPa | **1.45cpx = 0.02ol + 0.49opx + 1liq** | **1.47cpx = 0.02ol + 0.52opx + 1liq** | **1.51cpx = 0.04ol + 0.55opx + 1liq** |
| | 0.92opx + 0.08ol = 1liq | 0.92opx + 0.08ol = 1liq | 0.96opx + 0.04ol = 1liq |
| 3 GPa | - | - | 0.52cpx + 0.31 g + 0.17ol = 1liq |
| | **1.53cpx + 0.33 g + 0.15ol = 1.01opx + 1liq** | **1.59cpx + 0.34 g + 0.16ol = 1.09opx + 1liq** | **1.91cpx + 0.36 g + 0.12 ol = 1.39opx + 1liq** |
| | 0.49 g + 0.14ol + 0.37opx = 1liq | 0.51 g + 0.25ol + 0.38opx = 1liq | - |
| | 0.16ol + 0.84opx = 1liq | 0.16ol + 0.85opx = 1liq | 0.87opx + 0.13ol = 1liq |
| 4 GPa | 0.28 g + 0.48cpx + 0.24ol = 1liq | 0.28 g + 0.47cpx + 0.25ol = 1liq | |
| | **0.25 g + 1.75cpx + 0.24ol = 1.23opx + 1liq** | **0.25 g + 1.80opx + 0.24ol = 1.32opx + 1ol** | |
| | 0.26 g + 0.22ol + 0.51opx = 1liq | 0.29 g + 0.22ol + 0.49opx = 1liq | |
| | 0.22ol + 0.78opx = 1liq | 0.22ol + 0.78opx = 1liq | |
| 5 GPa | 0.23 g + 0.45cpx + 0.32ol = 1liq | 23 g + 0.44cpx + 0.33ol = 1liq | |
| | **0.19 g + 1.70cpx + 0.31ol = 1.20opx + 1liq** | **0.19 g + 1.26cpx + 0.32ol = 1.26opx + 1liq** | |
| | 0.24 g + 0.29ol + 0.47opx = 1liq | 0.21 g + 0.29ol + 0.49opx = 1liq | |
| | 0.30 g + 0.70ol = 1liq | - | |
| 6 GPa | 0.20 g + 0.42cpx + 0.39ol = 1liq | 0.20 g + 0.41opx + 0.39ol = 1liq | |
| | **0.15 g + 1.39cpx + 0.39ol = 0.94opx + 1liq** | **0.15 g + 1.42cpx + 0.40ol = 0.96opx + 1liq** | |
| | 0.23 g + 0.37ol + 0.40opx = 1liq | 0.21 g + 0.41opx + 0.37ol = 1liq | |
| | 0.25 g + 0.75ol = 1liq | 0.22 g + 0.78ol = 1liq | |

Bold indicates Incongruent reactions producing olivine or orthopyroxene.

petrogenesis need to be considered. The average $Cr_2O_3/Al_2O_3$ for garnet peridotites from all studied cratons is 0.4. When plotted against bulk Mg#, $Cr_2O_3/Al_2O_3$ ratios for peridotite suites are scattered and extend throughout the field bracketed by the 1–6 GPa residue evolution lines. For the Slave, Siberian, and Slave cratons, they even extend beyond the low pressure experimental bounds to «1 GPa (Fig. 3f), suggesting a role for an additional process and/or a non-pyrolitic starting composition.

**Existing models for modification of cratonic peridotite by melt-rock reaction.** One explanation for the skewed and multimodal distributions of olivine abundance in Archaean cratonic peridotites across the studied cratons is a two stage process, in which a baseline melt extraction event resulted in a normal olivine abundance of 76–82%, and a modification process that acted to either decrease (orthopyroxene-rich) or increase (olivine-rich) the olivine abundance about this value. Building on observations from peridotites in modern tectonic settings, a number of authors have attributed orthopyroxene-enrichment in refractory cratonic peridotites to melt-rock reaction superimposed on earlier extensive melt extraction. The models fall into two broad groups.

One prominent view is that the cratonic lithosphere represents the product of low pressure melting and subduction thickening. In this context, the melt-rock reaction has been implicated during the envisaged subduction-accretion mechanism, for example by silica enrichment from slab-derived hydrous subduction zone melts[15,17]. Mantle wedge peridotites are found as xenoliths from active fore- and back-arc volcanoes and in metamorphic belts from slab-mantle interface in southwest Japan, eastern China and the Solomon Islands. Arc peridotites from these settings share the $Al_2O_3$-poor and orthopyroxene-rich[7] nature with Archaean cratonic peridotite. In arc peridotites, relict olivine within orthopyroxene crystals and orthopyroxenite veins cross-cutting peridotite offer textural evidence for the formation of orthopyroxene at the expense of olivine as a result of interaction with slab-derived fluid or melt. However, unlike cratonic peridotite, arc peridotites are widely associated with clinopyroxene-rich residues (wehrlite) and are enriched in LILE elements derived from the subducting slab. Further differences are that arc peridotites are Fe-rich due to addition from slab-derived fluid or melt and they never reach the high Mg# seen in Archaean cratonic peridotite. It has been proposed that higher degrees of melting, and thus higher Mg#, were achieved in Archaean subduction zones, producing siliceous melt that migrated upwards and metasomatised the overlying lithosphere and leaving an ultradepleted dunite residue[47].

The second class of models views the cratonic lithosphere as the residue after one or more very hot mantle melting events. Parallels may therefore be drawn with modern mantle plume environments where melt-rock reaction also occurs. The effect of reaction between plume basalt and peridotite is recognised, for example from Ethiopia[48] and Kerguelen[49], where the occurrence of clinopyroxene-bearing pyroxenite veins within dunite and harzburgite xenoliths is attributed to the formation of orthopyroxene at the expense of olivine. Reaction with plume melts also results in lower Mg# in peridotite xenoliths[48,49]. Indirect evidence for the presence of pyroxenite within the mantle comes from the composition of erupted melts[50,51]. Partial melts of hybrid peridotite-pyroxenite sources have higher NiO and $SiO_2$ and lower CaO and MgO than melts of pure peridotite, on this basis it was suggested that pyroxenite contributes ~30% to ocean island basalts and ~60% to continental flood basalts[50]. Sobolev[50] attributed pyroxenite formation to peridotite interaction with melt derived from recycled, rather than asthenospheric basalt. Similar models involving melting of a hybrid peridotite-eclogite source have been suggested for the formation of the Archaean cratonic lithosphere[21,22].

Variations on these two types of models have also been proposed, most notably localised melt-rock reaction and silica enrichment that occurred sometime after craton stabilisation during local magmatic events[52]. Regardless of the mode of formation of the cratonic lithosphere and the timing of melt-rock reaction, an important constraint has recently been

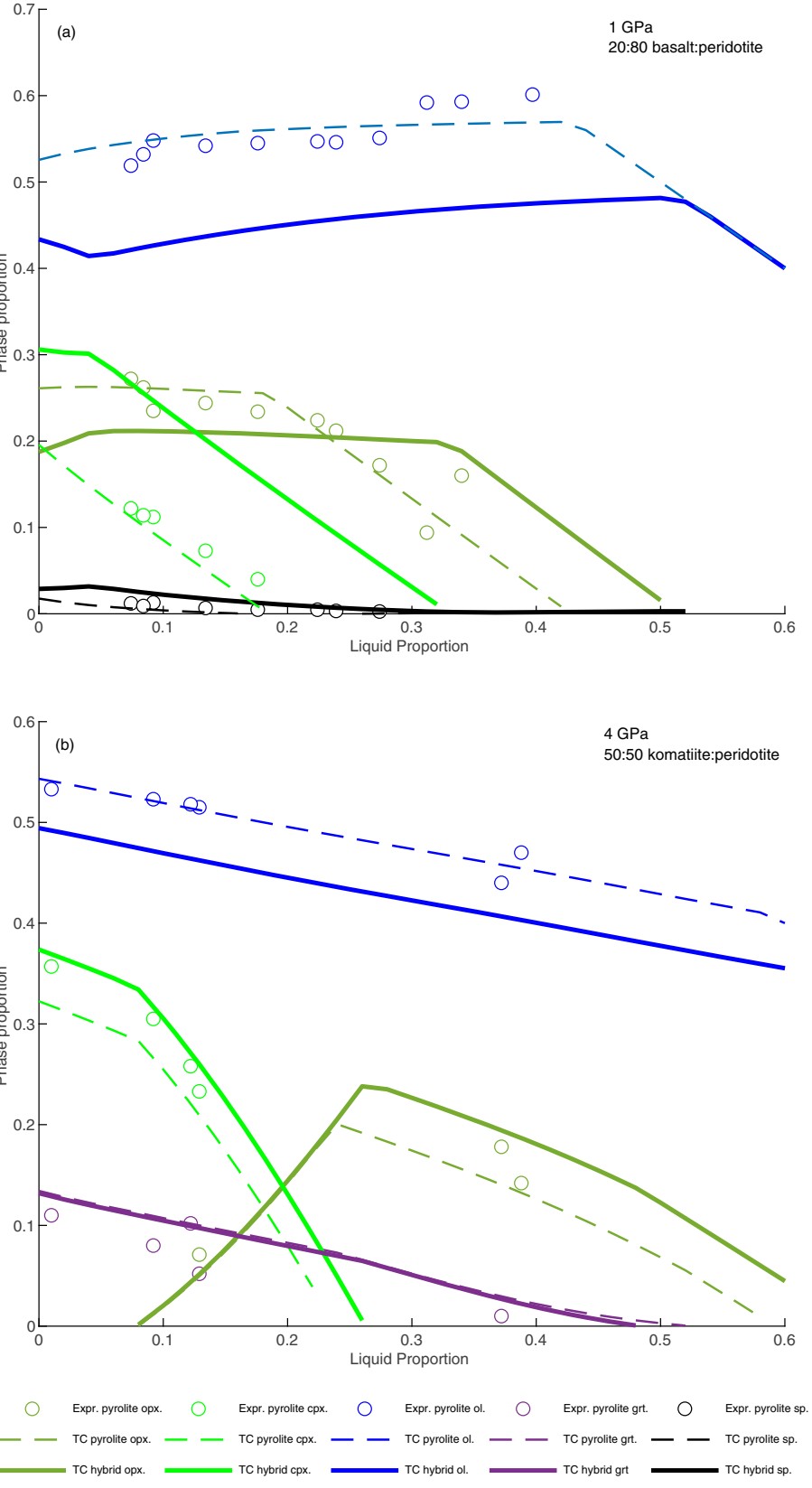

**Fig. 4 Mol. proportions of residual crystalline phases 'vs' liquid fraction in the modelled melting trends. a** 20:80 basalt:peridotite compared to fertile peridotite KR4003 at 1 GPa; experimental phase proportions for fertile peridotite MM3 are shown for comparison[43]; **b** 50:50 komatiite:peridotite compared to fertile peridotite KR4003 at 4GPa; experimental phase proportions for fertile peridotite KR4003 are shown for comparison (open symbols[41]).

imposed with a comprehensive study of $\delta^{18}O$ compositions of Archaean SCLM peridotites[53]. Peridotites from all five studied cratons span a very narrow range of $\delta^{18}O$ that is indistinguishable from ordinary upper mantle. Furthermore, olivine and orthopyroxene appear in isotopic equilibrium regardless of modal mineralogy, arguing against late addition of metasomatic Si and O. The proportion of slab melt required to produce the average orthopyroxene content of different cratons varies from 0.15 to 0.35, locally reaching 0.9 for orthopyroxene-rich samples[15]. The O-isotope models[53] show that such high proportions of slab melt would result in clear deviations in $\delta^{18}O$. Therefore, the mantle-like $\delta^{18}O$ composition of SCLM peridotites requires that the reacting melt is of mantle, rather than recycled origin[53]. This strongly argues against proposals for subducted slab- and sediment-derived melts[22,24]. The new $\delta^{18}O$ data for Archaean SCLM peridotites contrast with those of Proterozoic age that were originally used[13] to advance the serpentinite-dehydration model.

Multiple factors control the mineralogy of the melt-reacted peridotite mantle. Mixed and layered basalt-peridotite reaction experiments show that olivine is formed at the expense of orthopyroxene at low pressure (<1.2 GPa[44]), whereas orthopyroxene is formed at the expense of olivine at medium pressure (1–2 GPa[54]), and garnet is present in addition to orthopyroxene in the reaction zone at high pressure (≥3 GPa[55,56]). This places constraints on the minimum depth of silica enrichment via reaction with mafic melts. Preferential dissolution of pyroxene and precipitation of olivine leads to an increase in porosity and permeability[57], permitting further melt-rock reaction and potentially accelerating dunite formation. Conversely, orthopyroxenite reactive boundary layers have low porosity[54], preventing further reaction with the larger peridotite body. Therefore, pervasive orthopyroxene formation is likely to occur only at conditions significantly above the peridotite solidus[54]. Petrographic evidence for high porosity and melt-rock reaction within the cratonic lithosphere comes from orthopyroxene-rich veins in garnet harzburgites[52,58] strongly suggestive of partially preserved reactions.

**New modelling of melt-rock reactions.** We have modelled melt-peridotite reactions using THERMOCALC at 1–6 GPa (supplementary information). The compositions of Archaean cratonic peridotite offer some constraints on the possible nature of reacting melts. The melt must have had higher $SiO_2$ than fertile mantle in order to produce $SiO_2$-rich residues and melt must have been FeO-poor in order to yield the refractory compositions of cratonic peridotites. Cratonic peridotites are characterised by <2 wt% $Al_2O_3$, therefore melt-rock reaction either occurred at low pressure in the spinel stability field or the reacting melt had low $Al_2O_3$, in either case garnet would not have been stabilised in the residue. In the following we explore the potential effect of reactions between precursor peridotite and ascending tholeiitic basalt and komatiite, the two dominant volcanic rocks erupted onto Archaean greenstone belts[12].

The starting compositions for the precursor peridotite were fertile composition KR4003[41] and moderately and highly depleted peridotite compositions to emulate interaction with mantle that had already undergone melt extraction. The starting compositions of the ascending melts were experimentally derived olivine normative tholeiite with 13.9 wt.% MgO (run 21 at 2 GPa, Hirose and Kushiro 1993[59]); or komatiite with 35 wt.% MgO (run 78 at 8 GPa,[60]). THERMOCALC calculations were performed at 20 and 50% basalt/komatiite, and at 1 GPa increments from 1 to 3 GPa (basalt) and 1 to 6 GPa (komatiite). Thus, we envisage reaction between ascending melt and peridotite that is concurrently undergoing melting (without the local melt yet extracted).

Because all our models are fluid-absent and do not envisage advected crustal-origin melts, they predict that residues and melts have the canonical mantle O-isotope compositions[53].

**Basalt-peridotite reaction.** For basalt-peridotite reaction, the model results (Fig. 5) indicate that reaction between basalt and peridotite leads to a decrease in the MgO content of the residue, resulting in lower Mg# at all stages of residue evolution. The hybrid residue moves from lherzolite to harzburgite and finally dunite (Fig. 4a) as temperature increases, consistent with experimental observations[44,54–56]. Importantly, reaction melting strongly decreases the temperature of orthopyroxene exhaustion in the hybrid system (by ~50 °C at melt:rock ratios of 20:80) thereby making it possible to form pure dunites at lower temperature and at lower Mg# than is achieved during simple melting of fertile peridotite. At low temperature, reaction between basalt and fertile peridotite leads to the formation of pyroxene at the expense of olivine. The pyroxene content is 4% higher, and the olivine content 4% lower at the solidus in the 20:80 basalt:peridotite hybrid system than in peridotite alone. As temperature increases, incongruent melting of pyroxene forms additional olivine (Fig. 4a, Table 2) leading to an overall decrease in $SiO_2$ in the hybrid residue. Reaction of basalt with moderately depleted peridotite (Mg#89.9) reduces the magnitude, but does not eliminate the effect of basalt-peridotite reaction in driving a decrease in $SiO_2$ and Mg#, whereas reaction with highly depleted peridotite (Mg# 92.4) generates residues that resemble those produced by simple melting of fertile peridotite (Supplemental figure S3).

Addition of basalt to fertile peridotite increases the silica content of the hybrid melting residue, however the residue produced at low melt:rock ratios (20:80) does not approach the median silica content of Kaapvaal peridotites (Fig. 5a), whereas that of the basalt-depleted peridotite hybrids are lower still. At higher melt:rock ratios (50:50), basalt-peridotite reaction produces high-$SiO_2$ residues; however, this results from addition of clinopyroxene, rather than orthopyroxene. Furthermore, high melt:rock ratios supress the Mg# of the residue such that the hybrid residue does not overlap with orthopyroxene-rich peridotites from Kaapvaal. The formation of wehrlite is consistent with the experiments of Mitchell and Grove[44] who showed that wehrlite forms below the liquidus at 1–2 GPa and also consistent with the occurrence of clinopyroxene-rich rocks in modern low-pressure ridge, arc and plume settings in which basalt-peridotite reaction occurs. We suggest that this argues against reaction between peridotite and basalt having caused the silica enrichment observed in Archaean peridotites from the Kaapvaal craton and elsewhere.

Our calculations show that reaction between basalt and peridotite at modest melt-rock ratios can replicate the median compositions of olivine-rich peridotites with anomalously low $SiO_2$ and elevated $Cr_2O_3/Al_2O_3$ at Mg# <93 (Fig. 5). This effect would likely be magnified under open-system conditions[61]. Major element compositional variation observed among dunites can be explained by localised variations in the basalt:peridotite ratio. Ancient melt-rock reaction is suggested by the compositions of Fe-rich tholeiitic amphibolites from the Archaean Kolar Schist Belt[62]. Elsewhere, more recent basalt-peridotite reaction may have led to olivine enrichment, such as in the Tanzanian craton where the occurrence of FeO-rich dunites was attributed to interaction with asthenospheric melts during either Cenozoic rifting or older magmatic events[37]. Several geochemical and petrographic observations are consistent with a melt-rock reaction origin for some cratonic dunites: the forsterite contents of olivines from Wiedemann Fjord dunites have a negative skew to Mg# that is lower than

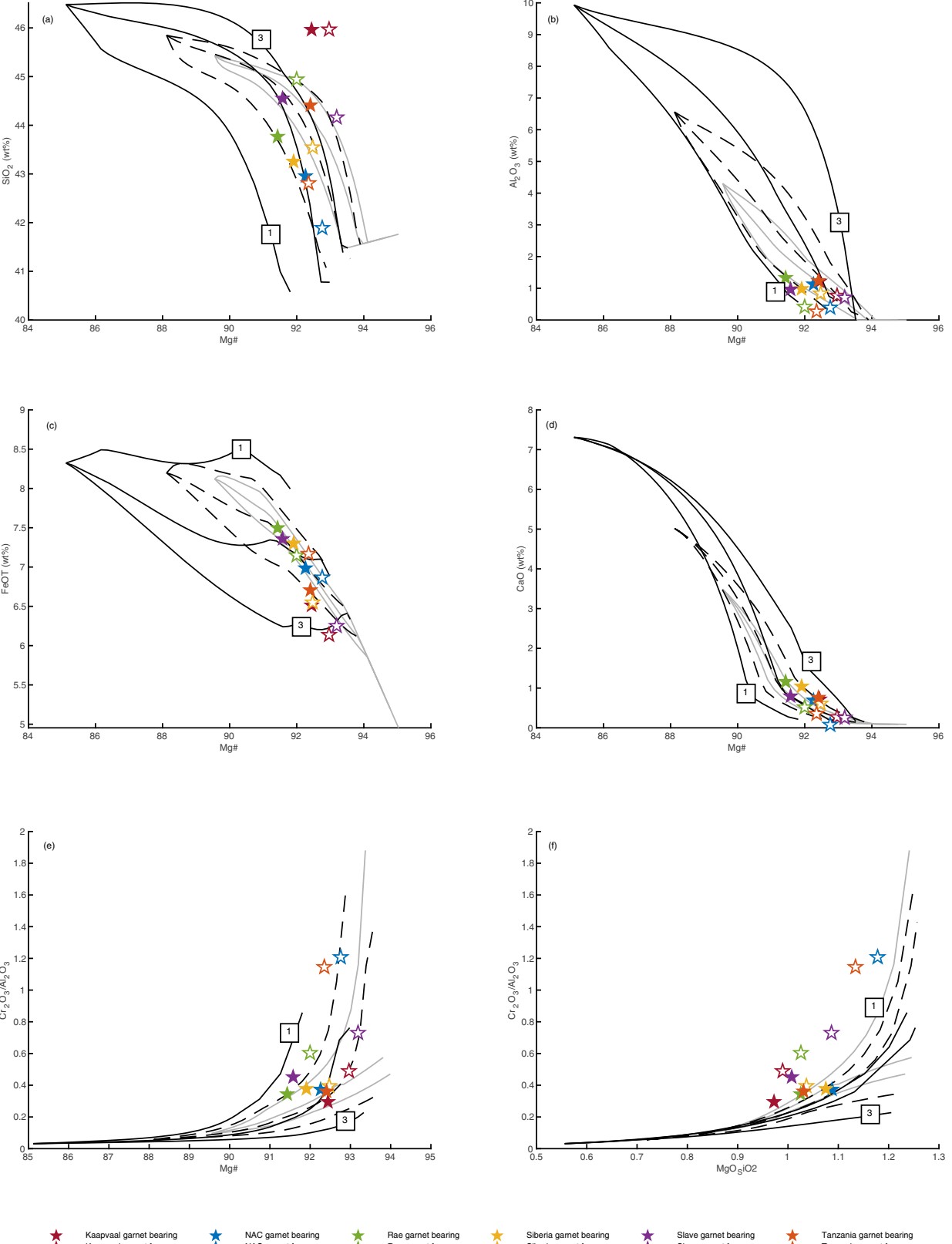

**Fig. 5 Modelled compositions residues after melting of basalt-peridotite hybrid rocks.** Starting compositions composed of 20% basalt and 80% fertile peridotite (black dashed lines) and 50% basalt and 50% fertile peridotite (black solid lines) compared to residues after melting of fertile peridotite (grey lines) from figure 3. (**a**) Mg# - SiO$_2$, (**b**) Mg# - Al$_2$O$_3$, (**c**) Mg# - FeO, (**d**) Mg# - CaO, (**e**) Mg# - Cr$_2$O$_3$/Al$_2$O$_3$, (**f**) MgO/SiO$_2$ - Cr$_2$O$_3$/Al$_2$O$_3$. Low and high pressure is labelled in GPa. Median compositions of garnet bearing (solid symbols) and garnet free (open symbols) cratonic peridotites were calculated from the literature database.

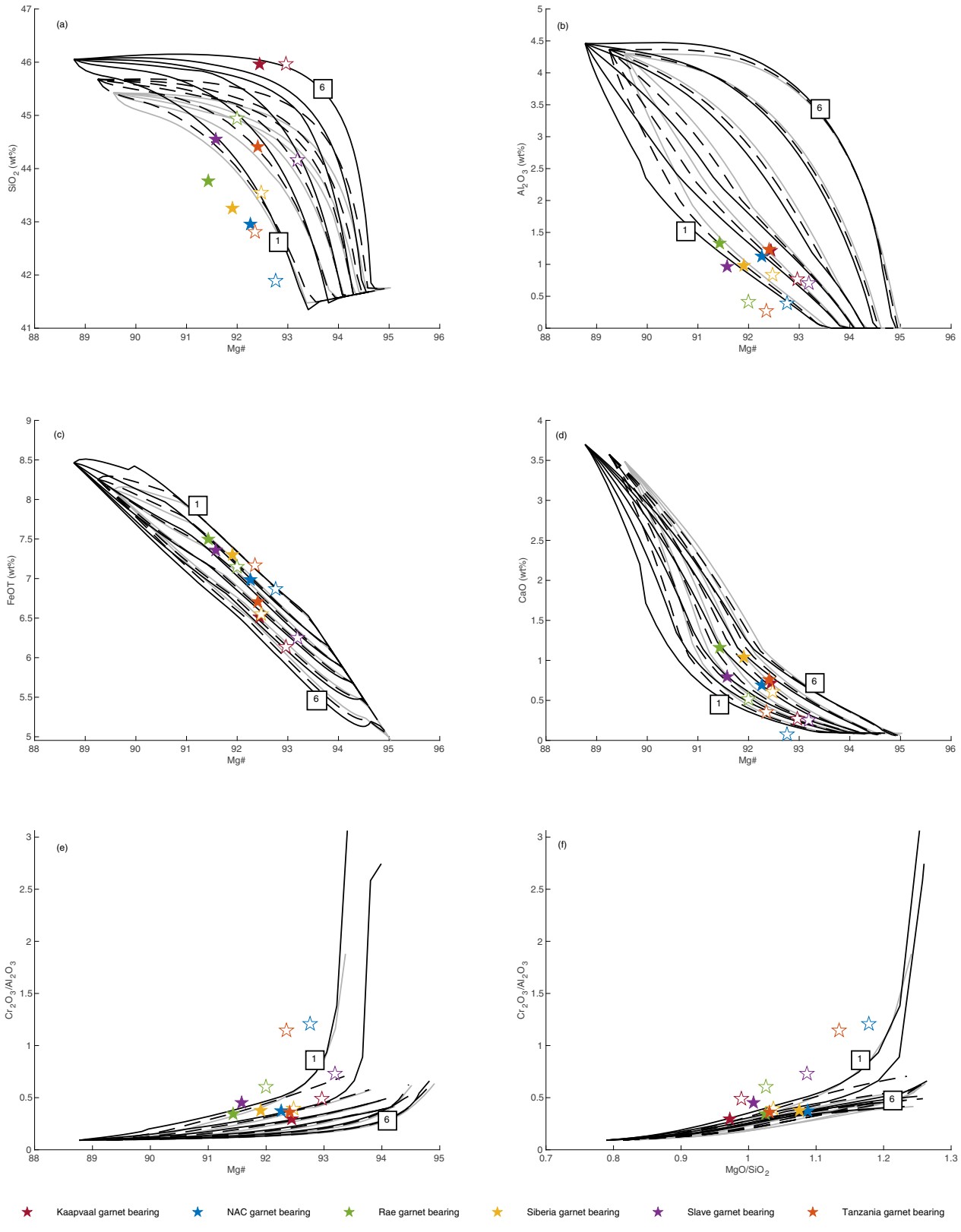

**Fig. 6 Modelled compositions residues after melting of komatiite-peridotite hybrid rocks composed of 20% komatiite and 80% fertile peridotite (black dashed lines) and 50% komatiite and 50% fertile peridotite (black solid lines) compared to residues after melting of fertile peridotite (grey lines) from figure 3.** (**a**) Mg# - $SiO_2$, (**b**) Mg# - $Al_2O_3$, (**c**) Mg# - FeO, (**d**) Mg# - CaO, (**e**) Mg# - $Cr_2O_3/Al_2O_3$, (**f**) $MgO/SiO_2$ - $Cr_2O_3/Al_2O_3$. Low and high pressures are labelled in GPa. Median compositions of garnet bearing (solid symbols) and garnet free (open symbols) cratonic peridotites were calculated from the literature database.

that of primitive mantle[5], whereas lithospheric dunites from Ubekendt Ejland coexist with cumulate dunites with low Mg#, and have higher bulk $Cr_2O_3$ and lower olivine NiO than observed in other cratonic suites[26].

**Komatiite-peridotite reaction.** For komatiite-peridotite reaction, the first overall result of the modelling is that reaction of komatiite melt with fertile peridotite increases the $SiO_2$ content of the melting residue at all pressures with minimal effect on the other oxides (Fig. 6). At low temperature, reaction between komatiite and fertile peridotite occurs in the orthopyroxene-free field and leads to the formation of clinopyroxene at the expense of olivine. The clinopyroxene content is 5% higher, and the olivine content 5% lower at the solidus in the 50:50 komatiite:peridotite hybrid system than in peridotite alone. As temperature increases, incongruent melting of clinopyroxene in the hybrid stabilises additional orthopyroxene in the residue (Table 2, Fig. 4b), leading to the observed increase in $SiO_2$. If melt-rock reaction takes place >50 °C above the peridotite solidus, orthopyroxene is formed directly and the residue is clinopyroxene-free. The $SiO_2$ content of the melting residue decreases at higher temperatures as orthopyroxene enters the melt, but remains high to Mg#≥93. The abundance of garnet in the hybrid komatiite-peridotite melting residue is fractionally lower than in the fertile peridotite melting residue for any given Mg#, but would be further reduced by reaction with komatiite of deeper origin. The effect of komatiite reaction with moderately depleted peridotite (Mg# 90.4) is virtually indistinguishable from that of reaction with fertile peridotite, whereas reaction of komatiite with highly depleted peridotite (Mg# 92.5) produces residues that resemble those produced by simple melting of fertile peridotite (Supplemental figure S4).

The median composition of garnet-bearing and garnet-free peridotites from the most silica-rich craton, Kaapvaal, can be modelled by progressive melting of a fertile peridotite-komatiite hybrid at high melt:rock ratios and at ≥4 GPa. Therefore, orthopyroxene-rich residues can be produced by melt-rock reaction at high komatiite:peridotite ratios. This effect may have been magnified under open-system conditions. The model cannot determine whether the addition of komatiite occurred as a single event or several events at lower melt:rock ratios. We attribute the wide range in $SiO_2$ among Kaapvaal peridotites to local variations in the overall komatiite:peridotite ratio and to bias from the large size of peritectic orthopyroxene formed during reaction melting. Melt-peridotite reaction is easier to identify when extreme, e.g., Kaapvaal craton. The diversity in $MgO/SiO_2$ of global cratonic peridotites is matched by the high $MgO/SiO_2$ variability of komatiites suggesting that neither represents the residue or liquid, respectively, of simple melting of pyrolite. Peritectic formation of orthopyroxene and new liquid from olivine and ascending reacting komatiite is the key reaction that leads to elevated orthopyroxene in the residue. Olivine inclusions in orthopyroxene may be important evidence for this reaction. The occurrence of olivine inclusions in garnet with higher Mg# than external olivine in xenoliths from Labait[37] and Lashaine[35] (Tanzania) is also suggestive of melt-rock reaction. Localised post-Archaean Si-enrichment may certainly have occurred but we envisage the main events to have been caused by Archaean komatiites because garnet that exsolved from very high-temperature orthopyroxene has Archaean-aged Lu/Hf isotope systematics[63]. Early episodes of komatiite-peridotite reaction may have pre-conditioned the SCLM for more Si-rich komatiite magmatism such as the Commondale suite at 3.3 Ga[64].

Peridotite-melt reaction during plume ascent has been discussed by Aulbach[22], who viewed orthopyroxene-enrichment as the product of reaction of peridotite with eclogite-derived melt, rather than komatiite, while Herzberg[10] suggested that

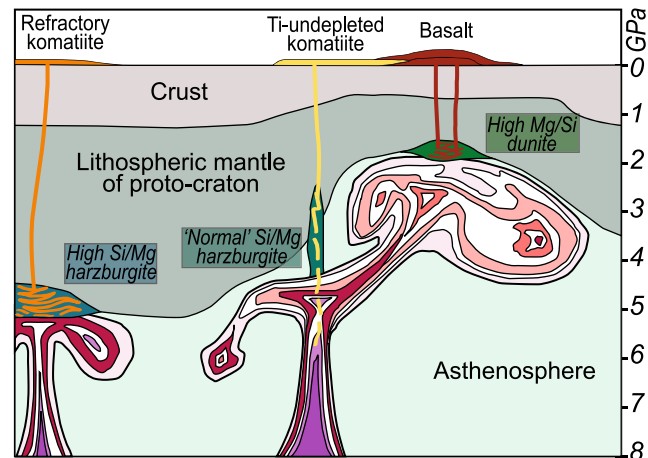

**Fig. 7 Schematic diagram showing three main mantle melting zones of a growing craton (vertically exaggerated).** The proto-cratonic lithosphere is thick on left-hand side but strongly extended on right-hand side. Two hot mantle upwellings are shown. On the left, the upwelling encounters strong refractory harzburgite and komatiite-harzburgite reaction forms new komatiite and residual orthopyroxene-rich, low $MgO/SiO_2$ harzburgite. In the centre, an upwelling asymmetrically impinges on the extended lithosphere. Within the root of the upwelling, polybaric melting of asthenosphere forms non-depleted komatiite. In the diverted expanding head, basalt is produced, reacting with pre-existing shallow harzburgite to form more basalt and residual high $MgO/SiO_2$ dunite.

orthopyroxene-rich peridotites may represent a mix of residual olivine and orthopyroxene formed as a cumulate from high-$SiO_2$ komatiite-like magma.

## Discussion

Figure 7 illustrates our working model for the growth of cratonic lithosphere, informed by statistical analysis of the composition of erupted mafic lavas;[12] the predominantly highly refractory (high Mg#) nature of Archaean cratonic harzburgites; and the comparison of the new modelling with observed peridotite mineralogies. At the heart of our model are a relatively cool baseline state of the mantle and the co-existence of two distinctive melting environments. The first is melting of relatively shallow mantle (<2 GPa) of modest potential temperature (50–100 °C above present) yielding the tholeiitic basalt piles that dominantly built Archaean greenstone belts. Interactions between the parental melts (MgO 14 wt%) of these basalts and shallow peridotite (both pyrolitic and harzburgitic) produced the variably olivine over-enriched residues that are found in the xenolith cargo from the shallower SCLM. In some locations, higher degrees of melting may have occurred leaving a refractory olivine-rich residue (Mg# ~93) such as represented by xenoliths at Wiedemann Fjord. The second melting environment is represented by relatively localised upwellings of much hotter mantle (1700–1800 °C) from greater depth (>4 GPa), erupting komatiite. Given the improbability of physically retaining a high melt fraction at these depths[65], we suggest that repeated interaction between komatiite and peridotite locally produced orthopyroxene-rich residues. Cooling after episodic and spatially variable thermal upwellings would have facilitated the survival of diamonds within the SCLM. Repeated hybrid melt extractions from different depths fit well with the eruption histories of greenstone belts and the diversity of komatiite chemistry.

The co-existence of shallow, modestly hot with deep, very hot melting throughout Earth history is evident in global geochemical igneous rock databases[66] and for the Archaean, is also supported

by greenstone belt petrology. Komatiite emplacement is spatially and temporally always associated with basalt. This suggests that advanced melting co-existed with relatively modest-degree melting. Where high-density data are available (e.g., Abitibi, Kalgoorlie, Barberton greenstone belts) there are no clear temporal trends in $Al_2O_3/TiO_2$ evolution of komatiites, i.e. there is no evidence for melting of progressively more refractory residues with time[67,68]. As a craton was building, and depleted residues formed, komatiite reaction would also have happened with harzburgite—not only pyrolite—as shown in Fig. 7, potentially yielding low-$TiO_2$ komatiites. Throughout the greenstone-building episodes, the growing residual mantle must have been at elevated temperatures and as melting progressed, the building residue of earlier melt extraction was in the way of following melts from deeper sources. Komatiite-peridotite interaction is thus an expected process.

On many cratons, mafic, and ultramafic magmatism occurred as relatively short-duration pulses (a few Ma) within overall periods of 50–150 Ma[69]. In our model, the operation of the two distinct melting regimes explains the wide $MgO/SiO_2$ distribution of cratonic peridotites as well as komatiites, which cannot be obtained by melt extraction from pyrolite alone. The extent of melting and melt-rock reaction is expected to have varied spatially and temporally leading to skewed and multimodal patterns of olivine abundance within suites of peridotites from a single locality and the wide range in $MgO/SiO_2$ of erupted mafic and ultramafic lavas.

The episodic upwelling of heat through a relatively cool background state mantle required by the model has two geodynamic implications. First, it necessitates the existence of a discrete strong thermal boundary layer well below the lithosphere. There is growing evidence that the base of the mantle transition zone could have been this layer and that many Archaean "plumes" originated from there[70]. The disappearance of the hypothesised layer would explain why both komatiites and coarse-grained orthopyroxene-rich harzburgites are largely absent after the Archaean. It would also have marked the onset of whole-mantle convection.

The second geodynamic constraint is the need for coexisting shallow cooler and deeper hot melting regimes. It is difficult to envisage these working within an already stabilised cratonic lithosphere, unless the early cratons were prone to pronounced thermal erosion, which could have permitted localised shallow melting (as envisaged in Fig. 7). An alternative, the more-speculative explanation is that cratonic crust and mantle may not strictly have formed together. The crust may have only been underlain by a shallow residue and petrologically independent residue zones may have existed at depth[12]. This could explain the lack of complementarity in highly lithophile elements between SCLM and cratonic crust and would allow the delamination of eclogite from the base of the differentiating crust. In both explanations, the very hot mantle upwellings, and the associated melt-rock reaction, are the distinguishing feature that significantly shaped the early continents.

### Data availability
The authors declare that the data (compiled literature data for cratonic peridotites and THERMOCALC output) are provided as supplementary data sets.

### Code availability
The THERMOCALC files necessary for performing the calculations may be found by following links to software at https://hpxeosandthermocalc.org/

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

## Acknowledgements

We would like to thank professor Tim Holland for developing the NCAFMASTOCr model for THERMOCALC and making it available for this study.

## Author contributions

E.L.T. and B.S.K. discussed and advanced the research ideas and collated the literature database. E.L.T. undertook the KDE and THERMOCALC modelling and wrote the paper. B.S.K. edited several iterations of the manuscript and created Fig. 7. E.L.T. and B.S.K. jointly revised the original submission.

## Competing interests

The authors declare no competing interests.
