## [Peer Review File · Nature Communications]

REVIEWER COMMENTS

Reviewer #1 (Remarks to the Author):

'Peridotite-komatiite interaction and the origin of variable silica in cratonic peridotite'

by E Tomlinson and B Kamber

This is an interesting and important manuscript exploring the evolution of ancient continental mantle lithosphere with focus on a key issue – the excess silica in cratonic peridotites. The authors set the scene nicely by explaining to the layman how cratons are thought to form and what mechanisms have previously been discussed to explain widespread silica enrichment. An updated database of cratonic peridotite major element compositions and modal mineral abundances for major cratons worldwide is presented, and the case of silica enrichment is reassessed using statistical methods. A major outcome of this exercise is that single-stage melting of primitive mantle cannot produce mantle residues that would resemble the silica-rich cratonic harzburgites with their skewed olivine modes. The authors use this first-order observation to develop one of the early ideas (early Herzberg works from the 1980-90s; cited) that komatiite melt interaction with depleted or fertile peridotite could potentially explain the compositions of many cratonic peridotite suites, and in particular those from the Kaapvaal craton in southern Africa. To this end the authors employ an updated version of the program Thermocalc for thermodynamic modeling of melt-rock reactions (a companion paper outlining the model is currently under review with JPET and has been provided to Nature Communications for evaluation).

I particularly enjoyed how this rather theoretical modeling is elegantly tied into the geology of Archean greenstone belts, accounting for several key features beyond silica-enrichment of cratonic peridotites, such as the proportions of tholeiitic basalts and komatiites at surface as well as the geochemical diversity of 'natural' komatiites. In my opinion, it is this combination that warrants publication in Nature Communications, because not many research teams would currently be able to tie all of this information so elegantly together (i.e., modelers hardly know field geology; field geologists are poor modelers; petrologists tend to get lost in detail; Tomlinson & Kamber master and fuse these disciplines convincingly).

Sebastian Tappe

30/09/2020

Johannesburg

I have a few minor comments that the authors may want to address:

- 1) The paper is perhaps a bit long and could be further streamlined with some of the modeling details being outsourced to an electronic appendix.
- 2) For southern Africa, the oldest diamonds within the cratonic mantle lithosphere coincide with major episodes of komatiite eruptions at surface (3.3-3.2 Ga). The presented thermodynamic model suggests that the komatiite melt - harzburgite rock reactions that produce excess orthopyroxene

occur at suprasolidus conditions (L328-329), so how can such diamonds survive or even form in the first place at these conditions?

3) Perhaps a little provocative: the Kaapvaal mantle is somewhat unique in terms of excess silica – and there is also a rather unique igneous event at ca. 2 Ga, the intrusion of the giant Bushveld Complex. There is growing evidence that the Bushveld igneous event was spatially even more widespread than what is currently exposed at surface (McCarthy et al., 2018 - SAJG). Some fairly established researchers argue that the parental magmas to the Bushveld Complex were komatiitic in character, i.e. very high-MgO coupled to high-SiO₂ (e.g., Maier et al., 2016 – CMP; Wilson, 2012 - JPET). I am wondering whether such ‘komatiitic’ magmas at around 2 Ga (actually a common komatiite magmatic episode around the globe) could have reacted with the Kaapvaal craton root to produce the widespread silica-enrichment? Preserved Archean Re-Os model ages mostly constrain the depletion history of olivine and sulphides but not necessarily tell about addition of “new” orthopyroxene. This idea still conforms with the constraint that the high-silica peridotites must have existed before 1.15 Ga, i.e. before the oldest kimberlite event that sampled these materials.

4) Chicken-and-egg: could the strong opx-enrichment precondition deep lithospheric mantle sources for high-MgO-high-SiO₂ komatiite volcanism such as the Comondale suite at 3.3 Ga (Wilson et al., 1989 - EPSL)?

5) The paper would benefit from a definition of ‘pyrolite’ because the term is a bit out of fashion. Maybe as a footnote or jargon box?

6) The statement in L315-317 about the apparent lack of recycled nitrogen in diamonds before 3 Ga is incorrect and needs clarification/discussion. The cited study (Howell et al., 2020) focused exclusively on inclusion-bearing diamonds with an ‘age’ attached. Obviously, this makes only a minute portion of diamonds present in the mantle (or even in our collections) and randomly sampled by kimberlite. Smart et al. (2016 – Nature Geoscience) showed clearly that crustal nitrogen contributed to diamond formation at >150 km depth before 3 Ga using firm evidence from the Witwatersrand placer diamonds. Their C-N isotope modeling suggests that up to 20% recycled sediment component contributed to the C-N budget of the mantle fluids that formed these diamonds in the root of the stabilizing Kaapvaal continent prior to 3 Ga. Also, Sobolev et al. (2019 – Nature) argued for recycled surface water to be present in the deep upper mantle sources to Kaapvaal craton komatiites by 3.3 Ga. Both these studies should be considered and the rather bold statement about “recycled material” should be refined.

7) The argument about mantle-like d¹⁸O values for cratonic peridotites and therefore a lack of crust-derived components in the cratonic mantle in the form of slab-derived melts/fluids is odd (L308-313). The d¹⁸O values were determined on high-Mg refractory olivine but not on newly added orthopyroxene, the subject of this current study. Also, to change the canonical mantle d¹⁸O value of mantle materials by crust contamination would require copious amounts of crust to be added to the mantle – meaning that small amounts of crust in the mantle are almost invisible for this technique.

8) Shouldn’t the updated peridotite database have information on PT for last equilibration?

9) Wasch et al. (2009 – GCA, cited) show that the garnet-opx clots in harzburgites at Kimberley have Nd-Hf isotopic compositions similar to the Mesozoic South African kimberlites. Couldn’t these coarse mineral assemblages just be metasomatic products and part of the megacryst formation event/s?

A few editorial suggestions and comments:

L1-2: The paper title could/should end with “...variable silica in cratonic mantle” (else you say

peridotite twice)

L57 and L250: it should be “Mg#” and “Cr#”

L76: dash missing?

L133: Later on, other localities in Greenland are mentioned too (e.g., L396). They are best introduced here. Perhaps update the database with data by Aulbach et al. (2017 – JPET) for West Greenland?

Line 147: it should say: “Sarfartoq kimberlite dykes”; this is not a single body or locale.

L229: Figure number missing?

L273-274: something got lost here / words are missing

L284, L296, and elsewhere: Use of “Mg#” does not require to say “value”

L296-302: The pyroxenite – OIB story seems to be off-topic and space could be saved here.

L308: remove the comma

L316: it must be 3 Ga

L313: ‘oceanic slabs’ are mostly mantle lithosphere with a small portion of oceanic crust. Slab is not equal to crust.

L331-332: also Bell et al. (2005 – CMP)?

L396: spelling of Pyramidefjeld. Also Aulbach et al. (2017, JPET)?

L404, L477, and elsewhere: spelling of Wiedemann

L457: full-stop missing

L458: et al.

L490-491: Although I tend to agree, for Barberton and other Kaapvaal greenstone remnants we see a crude evolution from Al-depleted at 3.55 Ga (Komati) to Al-enriched (Commendale, Weltevreden etc) at 3.3 Ga.

L508-509: This is unclear especially cause-and-effect. Perhaps rephrase? The opx-rich harzburgites are still present so how does this explain the disappearance of hot komatiite magmatism?

L511-521: The closing paragraph could be a whole lot stronger! Did you run out of steam?

Table 1: Wiedemann Fjord is located in East Greenland not SE (makes a big difference to people working in Greenland); Premier pipe is not located on the Kimberley Block (there are other small geographic glitches in the table).

Again, this is a neat study and I am looking forward to see it published in Nature Communications.

Sebastian Tappe

30/09/2020

Johannesburg

Reviewer #2 (Remarks to the Author):

This manuscript tackles a long- and outstanding problem central to the formation of the continents head-on in a way I have not seen and that is very welcome. The problem is centred on the composition of cratonic mantle and how that formed. Significant variations in cratonic peridotite xenoliths and erupted magmatic melts have long been known. Over the last few decades Re-depletion models ages have added the knowledge that the age of the cratonic lithosphere broadly matches that of the overlying crust. Thus, the two are coupled but how and why? Unfortunately, many groups have become distracted by the equally important question of whether plate tectonics was

operating at the time and so the mechanism of formation of the differing peridotite suites has fallen into the background. Thus, this paper is particularly welcome and very appropriate for Nature Communications because it is complementary to those studies concerned with when tectonics started.

The authors have undertaken a statistical and thermodynamic modelling approach which despite being very time consuming has the advantage of being unbiased. Assuming that the version of ThermoCalc does not contain flaws (and I very much doubt it does but I am not a thermodynamicist) then I think the results have to be considered robust and I recommend publication with minor revisions. The first of these is perhaps the hardest. Nature Communications is read by many non-specialists and I think the authors need to do a better job of letting a biologist or astronomer know why this is a big problem and why they should care about the results. The second concerns the reason why two melts formed at very different temperatures are produced at the same time. I don't disagree that they do but the explanation must have major geodynamic implications. The usual explanation for the komatiites come from plumes but I've not seen a convincing reason why in a convecting mantle plumes would have been so much hotter in the Archaean. Conversely, if they come from plumes what is the cause of the melting that produced the tholeiites? A final suggestion is that the authors make a clear statement about why they have taken the approach they have and not considered trace element or isotope data. This should be easy to do. I look forward to seeing the paper published.

Reviewer #3 (Remarks to the Author):

Recommend major revision, which must contain some exploration of oxygen isotope data on the veracity of the interpretation that has arisen from the author's modelling. In this manuscript the overview of previous modelling and interpretations reads more like a review paper, rather than as a concise summary 'setting the scene' for the author's own contribution. This content (lines 162-332) should be shortened, perhaps with details moved to a supplementary document.

From a model-based approach, the authors are proposing a refinement of a class of explanations for the compositional character and range of SCLM, via comparing model predictions for a small range of variables with observed xenolith compositions. This is clearly a constructive and useful exercise as a contribution to explore the unique nature of the SCLM. However, as with all modelling, there are caveats that should be explained fully, and explored in more detail, if necessary in supplementary text. Modelling is only true to the mathematical algorithms used, starting assumptions and the variables used. However, ultimately whether that model really emulates reality depends on if it is compatible with other information that was not incorporated into the model. Thus an independent set of data that might not explained by the model are the oxygen isotope signatures of SCLM whole rocks, garnets and olivines (e.g. as mentioned in reference #11). Such apparent pitfalls for the author's proposed interpretation needs to be explored explicitly in the Discussion. Also, some details of the modelling need to be explored in more detail (if necessary by supplementary text). Of particular importance would be the choice of the starting composition of the peridotite with which ascending komatiite and basalt melts interacted - how stable are the modelling outcomes across a

range of 'reasonable' peridotite compositions (e.g. CaO and Al₂O₃)?

Response to Main points by Reviewer #1:

1) The paper is perhaps a bit long and could be further streamlined with some of the modeling details being outsourced to an electronic appendix.

We agree that this section was long (echoed in Point 1 of Reviewer #3) . We have shortened the text by 50% and moved the model description to the supplementary information. The remaining text focuses on the new observation that dunitic peridotites are too Fe-rich to be residues of high degree melting (not been previously noted), and the discussion of the use of $\text{Cr}_2\text{O}_3/\text{Al}_2\text{O}_3$ for determining depth of melting. The latter is essential for readers to understand why we model both shallow and deep melting. We retained table 2 and figure 3 that show our model results as these are important for comparison with the modelled products of melt rock reaction and for discussing silica poor peridotites.

2) For southern Africa, the oldest diamonds within the cratonic mantle lithosphere coincide with major episodes of komatiite eruptions at surface (3.3-3.2 Ga). The presented thermodynamic model suggests that the komatiite melt - harzburgite rock reactions that produce excess orthopyroxene occur at suprasolidus conditions (L328-329), so how can such diamonds survive or even form in the first place at these conditions?

We agree that silicate minerals that formed during the komatiite-peridotite reaction process are expected to have ages corresponding to komatiite magmatism. With regard to formation or survival of diamond, there are two relevant points that are now addressed in the discussion. Firstly, our model for the cratonic lithosphere (Fig. 7) specifically shows that the background state of the cratonic lithosphere was cool and only episodically and regionally perturbed by very hot upwellings. Thus, shallower diamonds could form upon rapid (<100 Ma) conductive lithospheric cooling after komatiite interaction. Their survival was facilitated by the episodic and spatially variable nature of komatiite magmatism and melt rock reaction. Secondly, the temperature of the diamond-graphite transition is strongly pressure dependent. At 5 GPa, it occurs at c. 1250°C but at 6 GPa, the transition occurs 380°C hotter, at c. 1630°C at 6GPa. Thus, deep formation of diamond is not incompatible with suprasolidus conditions.

3) Perhaps a little provocative: the Kaapvaal mantle is somewhat unique in terms of excess silica – and there is also a rather unique igneous event at ca. 2 Ga, the intrusion of the giant Bushveld Complex I am wondering whether such [Bushveld-aged] ‘komatiitic’ magmas at around 2 Ga (actually a common komatiite magmatic episode around the globe) could have reacted with the Kaapvaal craton root to produce the widespread silica-enrichment? Preserved Archean Re-Os model ages mostly constrain the depletion history of olivine and sulphides but not necessarily tell about addition of “new” orthopyroxene.

One finding of newer studies of cratonic harzburgite (e.g. Regier et al. 2018) is that excess silica is not unique to the Kaapvaal Craton. Our careful statistical analysis clearly underlines this new aspect of our study, which is now stated explicitly. The reviewer then advances the provocative hypothesis that the orthopyroxene megacrysts could only be 2 Ga old. This seems very unlikely, because many of these crystals show exsolved garnets identical in chemistry to those dated by Lu/Hf to 3.3 Ga. The detailed trace element and isotopic arguments for a Mesoarchean age of these crystals were explained on p. 144 of a recent paper of ours - Kamber, B. S. & Tomlinson, E. L. Chem. Geol. 511, 123–151 (2019) – and we now refer more specifically to the relevant page number.

4) *Chicken-and-egg: could the strong opx-enrichment precondition deep lithospheric mantle sources for high-MgO-high-SiO₂ komatiite volcanism such as the Comondale suite at 3.3 Ga (Wilson et al., 1989 - EPSL)?*

This is certainly a possibility and has already been mentioned in a much more recent paper on Comondale komatiites - Wilson, A. H. The Late-Paleoarchean Ultra-Depleted Comondale Komatiites: Earth's Hottest Lavas and Consequences for Eruption. *J. Petrol.* 60, 1575–1620 (2019). In the revised m/s we now refer to Wilson (2019).

5) *The paper would benefit from a definition of 'pyrolite' because the term is a bit out of fashion. Maybe as a footnote or jargon box?*

Agreed, the term is now defined.

6) *The statement in L315-317 about the apparent lack of recycled nitrogen in diamonds before 3 Ga is incorrect and needs clarification/discussion. The cited study (Howell et al., 2020) focused exclusively on inclusion-bearing diamonds with an 'age' attached. Obviously, this makes only a minute portion of diamonds present in the mantle (or even in our collections) and randomly sampled by kimberlite. Smart et al. (2016 – Nature Geoscience) showed clearly that crustal nitrogen contributed to diamond formation at >150 km depth before 3 Ga using firm evidence from the Witwatersrand placer diamonds. Their C-N isotope modeling suggests that up to 20% recycled sediment component contributed to the C-N budget of the mantle fluids that formed these diamonds in the root of the stabilizing Kaapvaal continent prior to 3 Ga. Also, Sobolev et al. (2019 – Nature) argued for recycled surface water to be present in the deep upper mantle sources to Kaapvaal craton komatiites by 3.3 Ga. Both these studies should be considered and the rather bold statement about "recycled material" should be refined.*

With regard to the first point – the origin of nitrogen – we agree that our original account was inaccurate and had the potential do inadvertently confuse the message of our work. The key is that the melt-rock reaction process does not need the involvement of recycled material. The reviewer is correct that this observation alone is not evidence against crustal recycling. We have rephrased the relevant passage of text and removed all reference to nitrogen isotopes.

With respect to the reviewer's request to discuss the evidence presented by Sobolev et al. (2019) for recycled surface water to be present in the deep upper mantle sources, we point out that this is a highly contentious claim. Only a few months later, the co-author of Sobolev et al. (2019) who provided the komatiite samples (Wilson) published his own paper - Wilson, A. H. *J. Petrol.* 60, 1575–1620 (2019) - in which he showed that the water and other volatile enrichment was more likely related to seawater-melt reaction during the subaqueous emplacement of the komatiite.

Our model neither argues for nor against subduction processes and crustal recycling – topics that are widely discussed in the current literature. Rather than entering this debate, we toned down statements that could give the impression we were advocating against plate tectonics on the early Earth.

7) *The argument about mantle-like d¹⁸O values for cratonic peridotites and therefore a lack of crust-derived components in the cratonic mantle in the form of slab-derived melts/fluids is odd (L308-313). The d¹⁸O values were determined on high-Mg refractory olivine but not on newly added orthopyroxene, the subject of this current study. Also, to change the canonical mantle d¹⁸O value of mantle materials by crust contamination would require copious amounts of crust to be added to the mantle – meaning that small amounts of crust in the mantle are almost invisible for this technique.*

These are clearly two significant points that could question the validity of our model.

With regard to the claim that Regier et al. (2018) determined O-isotope values only on high-Mg refractory olivine but not on newly added orthopyroxene, we note that the reviewer is incorrect on two accounts. Firstly, Regier et al. (2018) measured $\delta^{18}\text{O}$ on both olivine (table S-1) and orthopyroxene (table S-2). Secondly, the two phases were found to be in isotopic equilibrium [p. 8]: “Additionally, cratonic mantle Opx from variably Si-enriched xenoliths (Table S-2; mean $\delta^{18}\text{O}$ of 5.74 ± 0.27 ‰) are in isotopic equilibrium with associated olivine (α enstatite–olivine at 1300 °C = ~ 0.5 ‰; Rosenbaum et al., 1994).”

With regard to the point that O-isotopes would be insensitive to small amounts of recycled crust, the point is that the MgO-SiO₂ systematics would require large amounts of recycled crust. Kelemen (1988) already showed that 15 to 35% crustal melt would be required to produce the average orthopyroxene modes seen in different cratons, locally reaching up to 90% melt for the peridotites with highest opx contents. Importantly, Regier et al. (2018) specifically tested this possibility by including analyses of $\delta^{18}\text{O}$ in strongly silica-enriched peridotites. They found them to be isotopically identical to ordinary peridotites, concluding [p. 6] “that the Si-enriched nature of some samples is unlikely to be related to slab melt infiltration”.

Thus, the data by Regier et al. (2018) fully support our modelling results. The quantitative models that underline these points are discussed in response to reviewer #3.

8) Shouldn't the updated peridotite database have information on PT for last equilibration?

We have gone to great length and care to compile and recalculate the purest form of the original data (mineral mode, rock chemistry, mineral chemistry). From these data (digitally provided as a spreadsheet), interested readers can easily calculate PT of last equilibration with geothermobarometers of their choice. However, the PT of last equilibration is not required in our model and we therefore prefer not to make an arbitrary choice of geothermobarometer to calculate parameters that are immaterial to our study.

9) Wasch et al. (2009 – GCA, cited) show that the garnet-opx clots in harzburgites at Kimberley have Nd-Hf isotopic compositions similar to the Mesozoic South African kimberlites. Couldn't these coarse mineral assemblages just be metasomatic products and part of the megacryst formation event/s?

We referred to this study because it is the only one that has photographically documented the garnet-opx clots, which could represent arrested melt blebs. We are aware of the Nd-Hf model age calculations by Wasch et al. (2009) and have recently obtained Lu/Hf mineral isochron dates on such clots that yield c. 3.2 Ga, supporting the suggestion that they could be ancient hybrid melt blebs. The Lu/Hf garnet data of Wasch et al. (2009) plot on the same isochron. However, since our Lu/Hf data are only currently being written up for publication and since the possibility of melt blebs is peripheral to our main model, we have deleted reference to them in the revised m/s.

Response to Main points by Reviewer #2:

1) Nature Communications is read by many non-specialists and I think the authors need to do a better job of letting a biologist or astronomer know why this is a big problem and why they should care about the results.

We have embraced this suggestion and broadened the introductory text to appeal to a wider readership.

2) The second concerns the reason why two melts formed at very different temperatures are produced at the same time. I don't disagree that they do but the explanation must have major geodynamic implications. The usual explanation for the komatiites come from plumes but I've not seen a convincing reason why in a convecting mantle, plumes would have been so much hotter in the Archaean. Conversely, if they come from plumes what is the cause of the melting that produced the tholeiites?

We agree that the coexistence of a cool and hot state of the mantle has major geodynamic implications. In the revised m/s we spell out the geodynamic implications more clearly and suggest solutions, including with reference to the latest model published since submission of the original m/s: Wyman, D.A. Komatiites from Mantle Transition Zone Plumes. *Frontiers in Earth Science* 8 (2020): 383.

Our study is only one in a series of papers that are collectively firming up evidence for this puzzling aspect of the early Earth. The first paper that enunciated the paradox and provided a geodynamic rationale was: Davies G.F. Episodic layering of the early mantle by the 'basalt barrier' mechanism. *EPSL* 275 (2008). Several additional studies have since elaborated on the mechanism of seeding very hot upwellings at the base of the mantle transition zone. Many of these have been reviewed in Kamber and Tomlinson (2019), to which we now also refer more explicitly.

3) A final suggestion is that the authors make a clear statement about why they have taken the approach they have and not considered trace element or isotope data.

We are grateful for this suggestion. In the revised m/s we now explain explicitly that suprasolidus melting involves complex pyroxenes for which suitable thermodynamic models are currently under construction (e.g. the companion m/s by Tomlinson&Hollan, J. *Petrol*). However, there are no relevant trace element partition coefficient data for such pyroxenes, rendering modelling based on them and their isotopes too speculative at this point.

Response to Main points by Reviewer #3:

1) In this manuscript the overview of previous modelling and interpretations reads more like a review paper, rather than as a concise summary 'setting the scene' for the author's own contribution. This content (lines 162-332) should be shortened.

This echoes point 1 by Reviewer #1 and has been addressed by shortening the relevant section by 50%.

2) However, ultimately whether that model really emulates reality depends on if it is compatible with other information that was not incorporated into the model. Thus an independent set of data that might not be explained by the model are the oxygen isotope signatures of SCLM whole rocks, garnets and olivines (e.g. as mentioned in reference #11). Such apparent pitfalls for the author's proposed interpretation needs to be explored explicitly in the Discussion.

At first read, the suggestion that the independent O-isotope data (O being the most abundant of the major elements) could not be explained by our model is a serious challenge. However, as already explained under point 7 of Reviewer #1, our model is actually supported by the O-isotope data by Regier et al. (2018). These authors analysed olivine and orthopyroxene in similar coarse-grained 'low-temperature' harzburgites from 5 cratons (Slave, Rae, North Atlantic, Kaapvaal, and Siberian Archean). Their key conclusion being [p. 9] that "in summary, the $\delta^{18}\text{O}$ homogeneity of a wide range of cratonic mantle peridotites demonstrates that their variably Si-enriched, sometimes subcalcic garnet-bearing nature, cannot have been inherited from protoliths that experienced seafloor serpentinisation (e.g., Schulze, 1986; Canil and Lee, 2009).

With regards to this concern of the reviewer, it is critical that the comprehensive study of Regier et al. (2018) has come to opposite conclusion to Canil and Lee (2009), because the latter study is reference #11 to which the reviewer appeals. Regier et al. (2018) offered several suggestions for the discrepancy, the most likely being that the rocks studied by Canil and Lee (2009) from the Colorado Plateau are Proterozoic and not Archean in age. In the revised m/s we now clearly differentiate between the two O-isotope studies and explain that our model agrees with the data from the demonstrably Archean peridotites from 5 cratons but not with the Colorado Plateau data.

Finally, with regard to the uniqueness of the solutions from our model, we also note that Regier et al. (2018) have already provided independent mass-balanced models (their Fig. 4) that show that neither the serpentinite model advocated by Canil and Lee (2009) nor the slab-flux model of Smart et al. (2016), referred to by Reviewer #1 under their point #6, are supported.

The straightforward prediction from our model, which is fluid-free, is that Si-enriched harzburgites should have the canonical mantle O-isotope composition, which is exactly what they have. This expectation is now much more clearly stated.

3) Also, some details of the modelling need to be explored in more detail (if necessary by supplementary text). Of particular importance would be the choice of the starting composition of the peridotite with which ascending komatiite and basalt melts interacted - how stable are the modelling outcomes across a range of 'reasonable' peridotite compositions (e.g. CaO and Al₂O₃)?

This very valid point by the reviewer made us realise that we had neglected to mention our analysis of the stability of the model outcomes. The information provided in the supplementary material now demonstrates that the model is remarkably insensitive to the starting peridotite composition, certainly across the range of bulk compositions discussed as being reasonable, and we allude to this in the main text.

Minor comments

L1-2: The paper title could/should end with "...variable silica in cratonic mantle" (else you say peridotite twice)

Changed.

L57 and L250: it should be "Mg#" and "Cr#"

Changed.

L76: dash missing?

Changed.

L133: Later on, other localities in Greenland are mentioned too (e.g., L396). They are best introduced here.

Added Pyramidejfeld, others are already mentioned

Perhaps update the database with data by Aulbach et al. (2017 – JPET) for West Greenland?

Unfortunately, this is not a usable data set because neither whole rock nor modal abundance data are provided.

Line 147: it should say: “Sarfartoq kimberlite dykes”; this is not a single body or locale.

Changed.

L229: Figure number missing?

Changed.

L273-274: something got lost here / words are missing

Missing words added.

L284, L296, and elsewhere: Use of “Mg#” does not require to say “value”

Changed throughout.

L296-302: The pyroxenite – OIB story seems to be off-topic and space could be saved here.

We maintain that this is relevant because it demonstrates melt-rock reaction in a plume setting and so is similar to the mechanism proposed here.

L308: remove the comma

Changed.

L316: it must be 3 Ga

Sentence has been removed.

L313: ‘oceanic slabs’ are mostly mantle lithosphere with a small portion of oceanic crust. Slab is not equal to crust.

Changed crustal to recycled, however, this section has also undergone a broader change in response to general comments from reviewers #1 and #3.

L331-332: also Bell et al. (2005 – CMP)?

Reference added.

L396: spelling of Pyramidejfeld.

Changed.

L404, L477, and elsewhere: spelling of Wiedemann

Changed all occurrences.

L457: full-stop missing

Agreed, changed.

L458: *et al.*

The reviewer is mistaken, both cited references are single author.

L490-491: *Although I tend to agree, for Barberton and other Kaapvaal greenstone remnants we see a crude evolution from Al-depleted at 3.55 Ga (Komati) to Al-enriched (Commendale, Weltevreden etc) at 3.3 Ga.*

As we have shown in earlier publications (e.g. Kamber&Tomlinson, 2019) this ‘crude evolution’ is largely an artefact of small datasets and breaks down when larger sample sets are included, which is why we state that this is “when high density data sets are available”.

L508-509: *This is unclear especially cause-and-effect. Perhaps rephrase? The opx-rich harzburgites are still present so how does this explain the disappearance of hot komatiite magmatism?*

We agree that this sentence was confusing and have re-phased it.

L511-521: *The closing paragraph could be a whole lot stronger!*

The revised introduction no links better to the closing paragraph. The closing statement about the first emerged landmasses is very strong in our view and should appeal to the broad readership of the journal.

Table 1: Wiedemann Fjord is located in East Greenland not SE (makes a big difference to people working in Greenland); Premier pipe is not located on the Kimberley Block (there are other small geographic glitches in the table).

Changed.

REVIEWERS' COMMENTS

Reviewer #1 (Remarks to the Author):

I have read the revised version and the authors are applauded for creating this well-argued and interesting paper, which will make a strong contribution to Nature Communications. Best wishes.
Sebastian Tappe

Reviewer #2 (Remarks to the Author):

I am happy that the authors have addressed my suggestions and suggest that this manuscript is now ready for publication.

Reviewer #3 (Remarks to the Author):

The authors have clearly taken on board the questions / comments undertaken on the first submitted version, and made significant changes to the manuscript that will improve the clarity of their methodologies (and why they have not undertaken other approaches), and the significance (and possible limitations) of their outcomes.

As such, as a scholarly document, I consider it is now in a form suitable for publication and that it is a subject suitable for Nature Comms. This is not to say that I agree with their 'universal' model to explain the enigmas of the cratonic mantle lithosphere. However, as now revised, they have a more robust case, which hopefully will foster debate from those interested in this issue.